# Multitask learning over shared subspaces

**Nicholas Menghi**\*, **Kemal Kacar**, **Will Penny**[ID]\*

School of Psychology, University of East Anglia, Norwich Research Park, Norwich, Norfolk, United Kingdom

\* n.menghi@uea.ac.uk (NB); w.penny@uea.ac.uk (WP)

## Abstract

This paper uses constructs from machine learning to define pairs of learning tasks that either shared or did not share a common subspace. Human subjects then learnt these tasks using a feedback-based approach and we hypothesised that learning would be boosted for shared subspaces. Our findings broadly supported this hypothesis with either better performance on the second task if it shared the same subspace as the first, or positive correlations over task performance for shared subspaces. These empirical findings were compared to the behaviour of a Neural Network model trained using sequential Bayesian learning and human performance was found to be consistent with a minimal capacity variant of this model. Networks with an increased representational capacity, and networks without Bayesian learning, did not show these transfer effects. We propose that the concept of shared subspaces provides a useful framework for the experimental study of human multitask and transfer learning.

**Data Availability Statement:** Behavioural data and Matlab (Mathworks Inc) software implementing the neural network models are available from https://github.com/wpennyUEA/subspace.

**Funding:** The authors received no specific funding for this work.

## Author summary

How does knowledge gained from previous experience affect learning of new tasks? This question of "Transfer Learning" has been addressed by teachers, psychologists, and more recently by researchers in the fields of neural networks and machine learning. Leveraging constructs from machine learning, we designed pairs of learning tasks that either shared or did not share a common subspace. We compared the dynamics of transfer learning in humans with those of a multitask neural network model, finding that human performance was consistent with a minimal capacity variant of the model. Learning was boosted in the second task if the same subspace was shared between tasks. Additionally, accuracy between tasks was positively correlated but only when they shared the same subspace. Our results highlight the roles of subspaces, showing how they could act as a learning boost if shared, and be detrimental if not.

## Introduction

Recent advances in machine learning have delivered human-like levels of performance across a variety of domains from speech and image recognition [1] to language understanding [2] and game-playing [3]. These advances have been achieved, in the main, using neural network models with very large numbers (e.g. millions) of parameters that are estimated using very large numbers (e.g. millions) of data points. The requirement for such a huge amount of

**Competing interests:** The authors have declared that no competing interests exist.

training data places limits on the tasks that can be learnt and is at odds with much of the psychology literature on human learning which suggests that concepts can be learnt using very few examples. One way of achieving such "data-efficient" learning is to leverage information learnt on one task to more efficiently learn another. Subfields of machine learning that have been using this approach include Multitask Learning (learning multiple tasks simultaneously) [4, 5], Transfer Learning (learning tasks sequentially) [6, 7] and Continual Learning [8, 9] (learning an indefinite number of tasks sequentially). This paper uses constructs from the machine learning literature to better understand how humans learn across multiple tasks.

Our starting point is the original Multitask Learning architecture proposed by Caruana et al. [4] in which generalisation across tasks is achieved using shared parameters. This architecture comprises a feature module, which can be shared across tasks, and an output module which is task-specific. In the original "hard-parameter sharing" architecture [5] the parameters defining the feature model are identical across tasks. Mathematically, this feature model defines a subspace that is shared across tasks. The idea that shared subspaces are useful for learning over multiple tasks has previously been highlighted, for example, under the term "structure learning" [10].

This paper uses an experimental design in which participants learn a pair of tasks that either do or do not share a common subspace. We investigate how learning proceeds with the hypothesis that learning will be facilitated for tasks that share a common subspace. Facilitation of learning could be manifested as faster and/or more accurate learning. We restrict ourselves to linear subspaces so that the shared features are a reduced-dimension linear projection of the input space, leaving nonlinear subspaces to subsequent experiments.

In additional modelling work we make use of a second construct from the Multitask Learning literature—that of "soft-parameter sharing" [5]. Here, a second task does not share exactly the same feature model, but parameters determining the features are constrained to be similar. We use a Sequential Bayesian learning algorithm for neural network training, also known as Elastic Weight Consolidation (EWC) [11], in which the prior over feature parameters for a second task is given by the posterior over feature parameters from the first. This is implemented by having two parameters for each network connection, a "mean" and a "precision", which together specify a Gaussian probability distribution. Bayesian estimation results in high precisions for those connections that have strongly adapted to data, and lower precisions for those that have not. Having a high precision makes connections more resilient to being overwritten on subsequent tasks. This is the mechanism for preventing so-called "catastrophic interference" (see [11] and [12] for discussion of potential neurobiological substrates). In this paper we use Sequential Bayesian learning over tasks and over mini-batches of data within a single task. This produces learning dynamics both within and between tasks, and the model predicts facilitation of learning (or "positive transfer" [13]) in tasks that share a common subspace. We compare these simulation results to empirical findings.

Overall, the paper presents a novel experimental task, empirical results on behavioural data, theoretical results from computer simulation, highlights similarities between them, and discusses ideas for future work in this area. We propose that the concept of shared subspaces provides a useful framework for the experimental study of human multitask and transfer learning.

## Materials and methods

### Ethics statement

All participants gave informed written consent, and the study procedure was approved by the local institutional review board of the University of East Anglia, UK. At the end of the experiment, participants received course credits for their participation.

## Notations

Our model-based analysis (see below) is described mathematically using the notation defined here. We use $N(x; m, \Lambda)$ to denote a multivariate Gaussian density over $x$ with mean $m$ and precision matrix $\Lambda$. The transpose of vector $x$ is written $x^T$. $1_{RK}$ denotes an $R$-by-$K$ matrix of ones, $A_{k\bullet}$ is the $k$th column of $A$ and $A_{\bullet k}$ is the $k$th row of $A$. The delta function $\Delta_{ab}$ takes the value 1 if $a = b$ and zero otherwise, $\mathrm{vec}(A)$ vectorises the matrix A into a column vector and the sigmoid function is given by

$$\sigma(x) = \frac{1}{1 + \exp(-x)} \tag{1}$$

## Participants

A total of ninety-six volunteers from the University of East Anglia (mean age = 19.90, SD = 1.36, 17 male) participated in the experiment. Data from seven participants became unavailable due to computer network synchronization errors. A further nine participants were discarded because they performed below chance level in both tasks. We performed our analysis on the remaining sample of 80 participants (mean age = 19.80, SD = 1.34, 13 male). All of them were naive to the purpose of the experiment.

## Apparatus and stimuli

The experiment was performed in a dimly lit room with participants seated 60 cm away from a computer display with their head supported by a chin-rest. Stimuli were presented on a 23-inch HP Elite Display 240c monitor using the Psychophysics Toolbox [14] for Matlab (Mathworks) running on Windows 7.

Two virtual "pies" (1° × 1° visual angle) were displayed at 1° from the central fixation point. Each pie was divided into six slices with from one up to five slices that could be filled with red colour, making a total of twenty-five combinations. The slices of the two pies were filled in a mirrored way as shown in Fig 1. The stimuli were presented on a dark grey background.

## Procedure

As we can see in Fig 1 each trial started with a black fixation cross presented at the center of the screen for an interval of 1000 ms. Afterwards, the stimuli appeared and stayed on screen

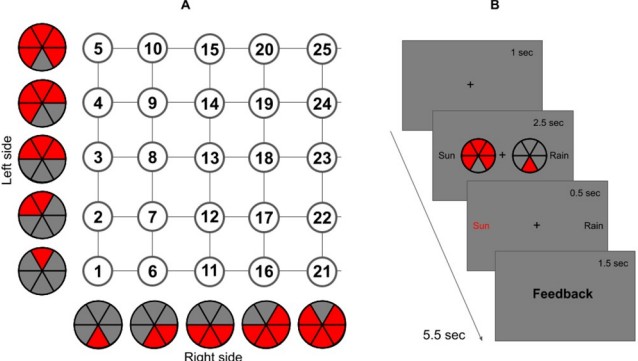

**Fig 1. (A) Experimental Stimuli. (B) Trial Structure**. (A) Each pie on the left/right side was combined with each pie on the right/left side, creating 25 potential configurations. (B) Each trial started with a fixation cross. Afterwards, two pies appeared and participants had up to 2.5 sec to respond. Confirmation of the choice was then given and feedback was provided.

for 2500 ms maximum or until a response was made. Responses were made on a standard keyboard, the "g" indicated sun/heads prediction in Task 1/2 and "j" indicated rain/tails in Task 1/2. Responses not given within the required time constitute "missed trials". Right after the button press, confirmation of the choice was given for 500 ms. Finally, feedback was provided, saying "correct" if the prediction was correct, "incorrect" if it was not and "too slow" for a missed trial.

The experiment took about one hour to complete and was composed of two tasks, comprising 250 trials each (10 repetitions per configuration) divided into 5 blocks, each of 50 trials. For the first task subjects had to make sun/rain decisions, as in the classic Weather Prediction Task [15], and for the second task they made heads/tails decisions ("Coin Prediction Task"). The mappings from stimulus to reward (correct/incorrect) were specified as described in the following section.

At the end of each task, we probed participants knowledge. We first asked them to describe the way they approached the task. We then gave them a list of six strategies (where only one was correct) and asked them to tick the one that resembled the most the one they used. Finally, we presented them with a timeline of the task asking to mark the point in time in which they started using that strategy.

## Stimulus-reward maps

Four different Stimulus-Reward Maps, or "Reward Functions", were used over the course of the experiment (but only two per subject), as shown in Fig 2. The underlying subspaces were operationalized by defining a common feature that, when represented, reduced the task to an approximate rule. In one case this was addition, in the other subtraction. Mathematically, the reward functions were generated using log-quadratic (Sub1, Add1) or log-linear (Sub2, Add2) mappings as follows. For the log-quadratic maps (Sub1, Add1), the probabilistic structure was

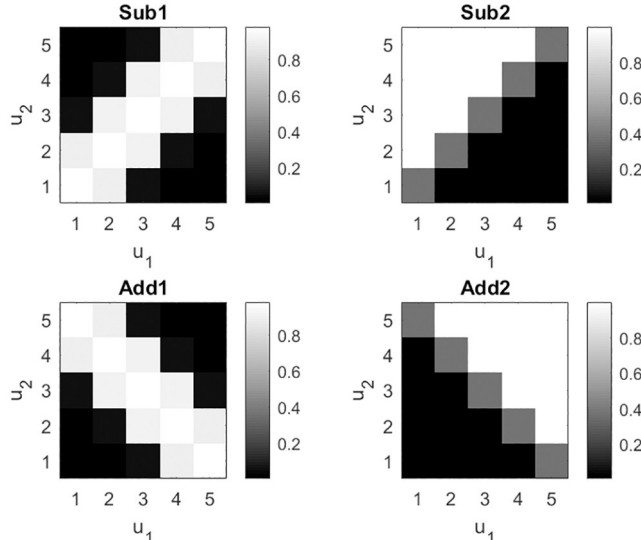

**Fig 2. Stimulus-reward maps.** Each gray scale image plots the reward probability (given button press "g" i.e. choosing Sun for Task 1 and Heads for Task 2) as a function of stimulus, $u$. The variables $u_1$ and $u_2$ denote the number of slices in the left and right pie stimuli, respectively. Each task can be implemented using two-stages of processing. For example, for the Sub1 and Sub2 maps the first stage requires extraction of a feature, $x = u_1 - u_2$. For Add1 and Add2 the required feature is $x = u_1 + u_2$. Tasks which use the same stimulus to feature space function (ie. subtraction or addition) are said to share the same subspace.

specified by making the log-odds of the outcome a quadratic function of stimulus characteristics. Flipping the sign of a single parameter in this mapping changes the Sub1 map to the Add1 map. That is

$$
\begin{aligned}
\log\left[\frac{p(y_t = 1)}{p(y_t = 0)}\right] &= (u_t - \mu)^T W (u_t - \mu) + w_0 \\
W &= 2.4 \times \begin{bmatrix} -0.71 & w_d \\ w_d & -0.71 \end{bmatrix} \\
\mu &= [3, 3]^T \\
w_0 &= 4
\end{aligned}
\tag{2}
$$

where $w_d = -0.71$ produces the Sub1 map and $w_d = 0.71$ produces the Add1 map. If, for each cue, subjects choose the option with the highest probability, then the correct classification rate would be 95 per cent. This is the maximum possible for the Sub1 and Add1 tasks.

We also defined tasks using a log-linear model which can produce, for example, the Sub2 map shown in Fig 2. Although generated from different models (log-linear versus log-quadratic), from a multitask learning perspective this task is similar to the Sub1 task in that the relevant feature for both tasks is $x = u_2 - u_1$ ie. subtraction. The Add2 map was similarly defined. The maximum performance levels for the Sub2 and Add2 maps were both 93 per cent.

Additionally, these maps could be approximately described using the following rules: Sub1—"Choose Sun if the difference in pie slices is zero"; Sub2—"Choose Heads if there more are slices on the left than right"; Add1—"Choose Sun if the sum of slices makes a full pie"; Add2—"Choose Heads if the sum of slices is greater than six".

## Experimental design

Each participant did two tasks, in the first one they had to learn the association between stimuli and weather outcome (sun or rain); in the second one they had to learn the association between stimuli and a coin toss outcome (heads or tails). Subjects were also explicitly instructed that the mapping in the second task was different. These two tasks were carried out on the same day in a 1 hour long experiment. The stimulus to outcome mapping in task 1 was specified by either the Sub1 or Add1 map. Task 2 was specified by either the Sub2 or Add2 map. Participants were assigned to either a "Same-Subspace" (Same) or "Different-Subspace" (Diff) group according to the logic of Table 1. There are 20 subjects per "condition" and 40 subjects per group.

Additionally, orthogonal subgroups of participants had a minimum 12 seconds break between one learning block and another whereas another orthogonal subgroup had minimum

**Table 1. Subjects and groups.**

| Condition | Task 1 | Task 2 | Subspace | Subjects |
|---|---|---|---|---|
| 1 | Add1 | Add2 | Same | 20 |
| 2 | Sub1 | Sub2 | Same | 20 |
| 3 | Add1 | Sub2 | Diff | 20 |
| 4 | Sub1 | Add2 | Diff | 20 |

Subjects were assigned to one of Same or Different Subspace Groups in a between-subjects design. Each of the Same/Different groups comprises data from two conditions e.g. data from the same subspace group is from both Add1-Add2 and Sub1-Sub2 conditions. There were 20 subjects assigned to each condition. For all subjects, Task 1 was presented as a weather prediction task and Task 2 as a coin prediction task.

120 seconds break between one block and another, in a two-by-two between-subject design (with factors of subspace and break-length). However, the break-length factor is ignored in the data analyses presented in this paper.

Given that participants are required to make Sun/Rain decisions and learn incrementally via feedback, Task 1 is reminiscent of the classic Weather Prediction Task (WPT) [16, 17]. However, a major difference is that in our tasks there is a hidden structure in the stimulus-reward mappings that can be discovered by subjects. Further, Task 1 is also similar to the Configural and Elemental Learning tasks defined by Duncan et al. [18], with elemental tasks containing a hidden structure (the log-odds of an outcome being a linearly separable function of stimuli). However, the hidden structure we have specified is a linear subspace lying within a non-linear (quadratic) mapping. Task 1 also shares similarities with the Feature-based Multi-Armed Bandit (FMAB) task of Stojic et al. [19] in that the reward probability is a function of bivariate stimuli. However, FMAB uses a linear function and participants make a multi-way (rather than binary) decision on each trial.

## Neural network model

This section describes a Neural Network model that we hope provides insight into some of the computational processes that may be engaged when solving Multitask learning problems. Learning in this model uses a sequential Bayesian estimation algorithm, similar to the Elastic Weight Consolidation approach [11], in which the prior over feature parameters for a second task is given by the posterior over feature parameters from the first. Bayesian learning for neural networks was first proposed by Mackay [20], and Bishop's textbook [21] provides a comprehensive introduction to the methodology. A novel aspect of our modelling work is that we implement sequential Bayesian learning over both tasks and mini-batches of trials within tasks, allowing the model to predict learning dynamics at the time scale of tens of trials. The neural network models are exposed to exactly the same stimuli and stimulus-reward maps provided to experimental participants, and in the results section we compare simulations from these models with empirical findings.

In the machine learning literature, Multitask Learning means training a neural network simultaneously on data with multiple output labels but where the inputs are of the same type, for example, learning to detect multiple types of object from the same visual images [4, 7]. Whereas, Transfer Learning means training a network sequentially on data from task A and then task B, but only tuning the final layer or layers using data from task B [6, 7]. We have designed our neural network model to accommodate both types of learning (using a mini-batch buffer to potentially store trials from multiple tasks) although our empirical data is from a transfer task.

Fig 3 shows our neural network model. It has a dynamic structure in which new output subnetworks are added as new tasks are encountered. New connections are created from units trained on previous tasks to units created for the new task. It is via these "Transfer Connections" that transfer of knowledge from one task to another is possible.

**Value network.** Let $r_t$ be a Bernoulli reward signal received after taking decision $d_t = k$ where $k = \{1, 2\}$ e.g. {Sun, Rain} in Task 1, and {Heads, Tails} in Task 2. A neural network is used to estimate the value (defined as the "expected reward" or "reward probability" [22]) to be obtained when choosing $k = 1, 2$

$$
\begin{aligned}
v_{t1}^n &\equiv p(r_t = 1 | d_t = 1) \\
v_{t2}^n &\equiv p(r_t = 1 | d_t = 2)
\end{aligned}
\tag{3}
$$

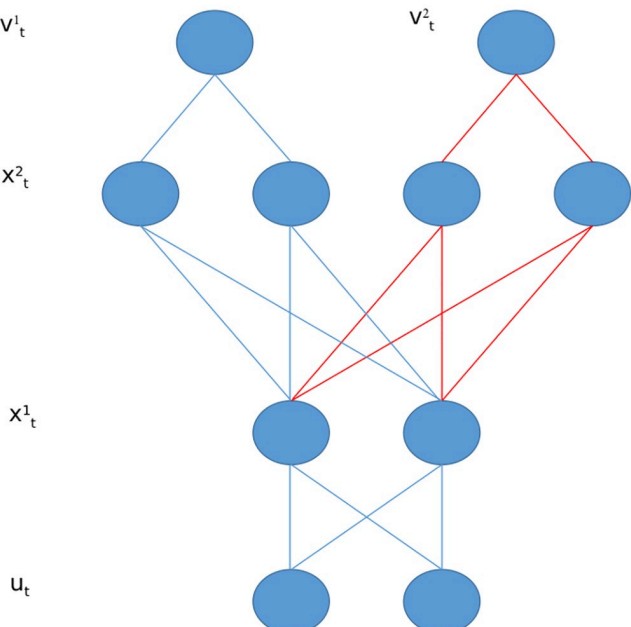

**Fig 3. Neural network architecture.** Sensory inputs, $u_t$ (where $t$ indexes trial number), map onto feature detectors in the first hidden layer, $x_t^1$, according to Eq 6. The corresponding weight matrix $W^1$ defines the feature subspace. Hidden units in a second hidden layer, $x_t^2$, further transform these (Eq 5) so that the output unit for the $n$th task, $v_t^n$, can provide task-specific value estimates for decision making (Eq 4). Here we depict two output networks, one for each task (weather prediction and coin prediction). Connections in blue exist when learning task 1 and are augmented by those in red when learning task 2. For the modelling results in this paper we used a *minimal capacity* network, having a single unit in the first hidden layer, and an *increased capacity* network having two units in the first hidden layer.

An artificial agent making decisions using these values takes a decision on trial $t$ for task $n$ by sampling from the Bernouilli distribution $v_t^n$. Here we assume that the task variable $s_t = n$ is known (i.e. agent performs task $s_t$ on trial $t$), that is, we have no task ambiguity.

We start our description of the neural net model at the output stage (top of Fig 3). In what follows $x$ variables denote the hidden unit output values, $w$ and $W$ connection strengths, $b$ biases, and $a$ the activations before entering the activation function that produces the output of each node. Superscripts 1, 2 and n denote first and second hidden layers and $n$th output sub-network. For each of $n = 1..N$ output subnetworks we have

$$
\begin{aligned}
v_{t1}^n &= \sigma(\tilde{a}_t^n) \\
v_{t2}^n &= 1 - \sigma(\tilde{a}_t^n) \\
\tilde{a}_{tk}^n &= \sum_{j=1}^{H} \tilde{w}_j^n \tilde{x}_{tj}^n + \tilde{b}^n \\
\tilde{x}_t^n &= P_n x_t^2
\end{aligned}
\tag{4}
$$

where $P_n$ is a selection matrix which selects those $H$ second layer units that belong to the $n$th output subnetwork. As our experimental paradigm involves binary decisions the above formulation with sigmoid functions suffices. More generally, with $K > 2$ potential actions, as with multi-armed bandits, values would need to be defined using softmax functions [21]. For $k = 1..$

$H_2$ nodes in the second hidden layer we have

$$
\begin{aligned}
x_{tk}^2 &= f_2(a_{tk}^2) \\
a_{tk}^2 &= \sum_{j=1}^{H_1} W_{kj}^2 x_{tj}^1 + b_k^2
\end{aligned}
\tag{5}
$$

where $f_2()$ is the activation function of the second-layer units and $W_k^2$ specifies the dependency of the layer two hidden units on the layer one units. This reflects the structure shown in Fig 3. For $k = 1..H_1$ nodes in the first hidden layer we have

$$
\begin{aligned}
x_{tk}^1 &= f_1(a_{tk}^1) \\
a_{tk}^1 &= \sum_{j=1}^{D} W_{kj}^1 u_{tj} + b_k^1
\end{aligned}
\tag{6}
$$

where $f_1()$ is the activation function of the first-layer units, and $D$ is the dimension of the input vector (i.e. number of inputs). A number of choices are available for the activation functions including Gaussian Error Linear Units (GELUs), $f(x) = x\Phi(x)$ where $\Phi$ is the Cumulative Density Function of the Gaussian distribution, Rectified Linear Units (RELUs), $f(x) = \max(0, x)$, Cosine Units, $f(x) = \cos(x)$ and linear units, $f(x) = x$. See [6] for a discussion of their relative merits. In this paper, for the first hidden layer we use linear units, and for the second hidden layer we use GELU units for Task 1 (as the mapping to output is nonlinear) and linear units for Task 2 (as the mapping to output is linear).

We augmented the inputs with a third input, $u_{t3} = 6$, reflecting the maximum number of slices in a pie, a variable readily available to human subjects. Thus we have $D = 3$ and the Add subspace can be represented with the weights $W_{k\bullet}^1 = [1, 1, -1]^T$ (sum of number of slices is maximal) and the Sub subspace with the weights $W_{k\bullet}^1 = [1, -1, 0]^T$ (difference in number of slices is zero).

We write the weights and biases that parameterise the neural network as $\{W, b\}$. Optimisation and statistical inference on these parameters is best described (and implemented in generic code) by first transforming them into a vector format [23]. We write this transformation generically as $\theta = \texttt{Pack}[W, b]$. For example, given a single task this $\texttt{Pack}$ function is

$$
\theta = [\texttt{vec}(W^1); \texttt{vec}(W^2); \tilde{w}^1; b^1; b^2; \tilde{b}^1]
\tag{7}
$$

Given two tasks we have

$$
\theta = [\texttt{vec}(W^1); \texttt{vec}(W^2); \tilde{w}^1; \tilde{w}^2; b^1; b^2; \tilde{b}^1; \tilde{b}^2]
\tag{8}
$$

After parameter estimation, we use the $\texttt{UnPack}$ function to recover $\{W, b\}$.

**Sequential Bayesian learning.** We update model parameters not after each trial, but rather after a "mini-batch" or "block" of training trials. In this paper we use Sequential Bayesian learning (SBL) over tasks where separate blocks contain data from different tasks, and over blocks of learning trials within each task. We define the $j$th block of training data, $R_j$, to comprise the input and task variables along with the decisions made by an agent and the rewards received. We write this as $R_j = \{r_t, d_t, s_t, u_t\}$ for $t \in \tau_j$ where $\tau_j$ is the set of all trials in the $j$th block.

In this paper, once a block of training data has been used for offline learning it is then discarded. To make best use of this data we use SBL so that information is efficiently propagated from one block to the next. We define $Y_j$ to denote all blocks of data up to and including block $j$. That is $Y_j = \{R_1, R_2, \ldots, R_j\}$. Bayesian estimation of $\theta$ proceeds over blocks such that the prior

over $\theta$ is updated to a posterior using Bayes rule

$$p(\theta|Y_j) = \frac{p(R_j|\theta)p(\theta|Y_{j-1})}{p(R_j)} \tag{9}$$

The likelihood of the $j$th block of data, $p(R_j|\theta)$, is defined in the following section (on "Model Likelihood"). We use a Laplace approximation to compute the posterior density, $p(\theta|Y_j)$, (see "Posterior Distribution" section below) which does not require explicit computation of the denominator term $p(R_j)$. We use a Gaussian prior over $\theta$ that factorises over parameters

$$p(\theta|Y_{j-1}) \quad = \quad \prod_{i=1}^{P} \mathbb{N}(\theta_i; m_{j-1}(i), \lambda_{j-1}(i)) \tag{10}$$

where $m_{j-1}$ is the prior mean, $\lambda_{j-1}$ is the prior precision and $P$ is the number of network parameters. For the first learning episode on the first task the prior is initialised with mean, $m_{j-1} = 0_P$, and prior precision $\lambda_{j-1}$ set so that hidden units with more inputs have smaller weights [23]. Given data $R_j$ from the first learning episode, SBL is used to compute the posterior distribution. This is also chosen to factorise over parameters

$$
\begin{aligned}
m_j &= \mathrm{MAP}(R_j, m_{j-1}, \lambda_{j-1}) \\
\lambda_j(i) &= \lambda_{j-1}(i) + \sum_{t \in \tau} \nu_{tk}(1 - \nu_{tk})\eta_t(i)^2 \\
\eta_t(i) &\equiv \frac{d\tilde{a}_t^n}{d\theta_i}
\end{aligned}
\tag{11}
$$

Here MAP refers to a gradient-based offline algorithm (see "MAP Estimation" section below) that finds the maximum-a-posterior parameters. That is, the parameters that are a-posteriori most likely. A fully factorised Laplace approximation is used to estimate the posterior precisions (see "Posterior Distribution" section below). The quantity $\eta_t(i)$ is referred to as the "output sensitivity" (the variable $\tilde{a}_t^n$ produces the network output as shown in Eq 4). Intuitively, network parameters $\theta_i$ that cause larger changes in the output will be better determined by the data and so be estimated more precisely.

The following sections on "Model Likelihood", "Prior Distribution", "Joint Distribution", "MAP Estimation" and "Posterior Distribution" break down each of the above steps into more detail, but can be skipped if technical details are not of interest.

In SBL, as with all dynamic Bayesian models (such as the HMM or Kalman Filter), the posterior from one learning episode becomes the prior for the next, as shown by Eq 9 which is applied recursively. If we were working with linear Gaussian models then SBL over $J$ mini-batches would be exactly equivalent to Bayesian learning from a single batch (comprising all exemplars) [21]. However, as we are using a fully factorised Laplace approximation in a non-linear model, its an empirical matter as to whether this procedure works well. The SBL approach, also known as Elastic Weight Consolidation (EWC), has previously been used for multitask learning of high-dimensional pattern recognition problems in the machine learning literature [11]. Here we apply SBL over mini-batches of data, as well as over tasks.

**Model likelihood.** Let $r_t$ be a Bernoulli reward signal received after taking action $d_t = k$. This paper employs an offline learning approach (similar in concept to offline Reinforcement Learning [24]), in which data is stored in a memory buffer. This buffer contains all inputs observed, task variables specified, decisions made and rewards received over a given set of

trials, $R_j = \{u_t, s_t, d_t, r_t\}$. The likelihood over the $j$th batch of data is then given by

$$
\begin{aligned}
p(R_j|\theta) &= \prod_{t\in\tau_j} p(r_t|d_t, s_t, u_t) \\
p(r_t|d_t, s_t, u_t) &= [v_{tk}^n]^{r_t}[1 - v_{tk}^n]^{(1-r_t)}
\end{aligned}
\tag{12}
$$

where $k = d_t$ is the selected action, $n = s_t$ is the selected task, $u_t$ is the sensory input on trial $t$, and $v_{tk}^n$ is the output of the value network. The Log Likelihood is

$$
\begin{aligned}
\log p(R_j|\theta) &= \sum_{t\in\tau_j} L_t \\
L_t &= r_t \log v_{tk}^n + (1 - r_t) \log (1 - v_{tk}^n)
\end{aligned}
\tag{13}
$$

We refer to the quantity $L_t$ as the sample log likelihood as it is based on a single data sample. The gradient, $g_t$, of the sample log likelihood is derived in the Supporting Information and computed using backpropagation. The Hessian (curvature) matrix is given by

$$
H(i, i') = \frac{d^2 \log p(R_j|\theta)}{d\theta_i d\theta_{i'}}
\tag{14}
$$

As in [11] we compute the Hessian using an outer-product approximation [25].

$$
\begin{aligned}
H &= -\sum_{t\in\tau_j} v_{tk}^n(1 - v_{tk}^n)\eta_t\eta_t^T \\
\eta_t(i) &= \frac{d\tilde{a}_t^n}{d\theta_i}
\end{aligned}
\tag{15}
$$

where $k$ and $n$ index the decisions made and tasks undertaken on trial $t$. The output sensitivity, $\eta_t$, can be computed using back-propagation (see S1 Text).

**Prior distribution.** The log prior is given by

$$
\begin{aligned}
\log p(\theta|Y_{j-1}) &= \sum_{i=1}^{P} \log \mathrm{N}(\theta_i; m_{j-1}(i), \lambda_{j-1}(i)) \\
&= -\frac{P}{2}\log(2\pi) + \frac{1}{2}\sum_{i=1}^{P}(\log \lambda_{j-1}(i) - \lambda_{j-1}(i)[\theta_i - m_{j-1}(i)]^2)
\end{aligned}
\tag{16}
$$

with gradient and curvature given by

$$
\begin{aligned}
\frac{d \log p(\theta)}{d\theta_i} &= -\lambda_{j-1}(i)[\theta_i - m_{j-1}(i)] \\
\frac{d^2 \log p(\theta)}{d\theta_i^2} &= -\lambda_{j-1}(i)
\end{aligned}
\tag{17}
$$

**Joint distribution.** We can then define the log joint density and its gradient as

$$
\begin{aligned}
J &= \log p(R_j|\theta) + \log p(\theta|Y_{j-1}) \\
\frac{dJ}{d\theta_i} &= \frac{d \log p(R_j|\theta)}{d\theta_i} + \frac{d \log p(\theta)}{d\theta_i} \\
&= \left( \sum_{t \in \tau_j} g_t \right) - \lambda_{j-1}(i)[\theta_i - m_{j-1}(i)]
\end{aligned}
\tag{18}
$$

where $g_t$ is the gradient of the sample log likelihood derived in the Supporting Information and computed using backpropagation. Bayesian learning from data set $R_j$ can then proceed by ascending the gradient of the log joint to reach a local maximum of the posterior density. Inclusion of the prior term ensures that parameter estimates are constrained to be similar to values found useful for previous blocks of data or for previous tasks (see last term in above equation).

Importantly, the prior precision $\lambda_{j-1}$ controls the strength of this effect, and this quantity increases in proportion to the number of data samples so far observed (in sequential Bayesian Learning for linear Gaussian models the posterior precision equals the prior precision plus the data precision and therefore always increases—See "Posterior Distribution" in S1 Text). This leads to the desirable property that the connection parameters converge to high precision solutions and is the mechanism described by Kirkpatrick et al. [11] for protecting previously learnt representations.

**MAP estimation.** Offline learning proceeds using gradient ascent. For the implementation in this paper, rather than using fixed step size updates we use a line search algorithm [26]. Specifically, on iteration $it$ of batch learning we use

$$
\theta(it + 1) = \theta(it) + \alpha \frac{dJ}{d\theta}
\tag{19}
$$

where $\frac{dJ}{d\theta}$ is the gradient of the log-joint. Optimal values for $\alpha$ are found using a single-variable bounded nonlinear function minimisation (implemented using fminbnd.m in Matlab (Mathworks, Inc) with step sizes bounded between 0 and 1) to minimise the negative Log Joint. If the above does not result in a decreased cost function (increased Log Joint) the maximum step size is reduced by a half and the process repeated. This can occur for a further three halvings of the maximal step size.

All $\theta$ values are initialised by sampling from the prior. This is a stochastic process which leads to different results on each simulation run. Other than this sampling process, the optimisation is deterministic. Additionally, we found that the posterior landscape contains local maxima. We therefore implemented a multistart optimisation procedure in which optimisation is re-initialised (with a different sample from the prior) until a satisfactory solution was found (see e.g. [27] for alternative multistart approaches). This was defined as a solution with an average trial likelihood of at least $pc_T = 0.60$. This is computed by dividing the log likelihood (Eq 13) by the number of trials and then exponentiating, and is also equivalent to the average probability of being correct [28]. If no such solution is found within a maximum of maxstarts starts the best solution is returned. For the results in this paper we used maxstarts = 3, the motivation for which is described in the results section.

**Posterior distribution.** We compute the posterior distribution over $\theta$ given data, $Y_j$, from blocks 1 to j. We use an approximate posterior based on a factorised Laplace approximation

$$
\begin{aligned}
p(\theta|Y_j) &= \prod_{i}^{P} p(\theta_i|Y_j) \\
p(\theta_i|Y_j) &= \mathrm{N}(\theta_i; m_j(i), \lambda_j(i))
\end{aligned}
\tag{20}
$$

where the posterior mean $m_j$ is the MAP estimate of $\theta$ found on the jth block of data (see above section). From the Laplace approximation we have

$$
\begin{aligned}
\lambda_j(i) &= \Lambda_j(i, i) \\
\Lambda_j &= -H + \Lambda_{j-1}
\end{aligned}
\tag{21}
$$

where the posterior precision, $\Lambda_j$, is the sum of the data precision ($-H$) and the prior precision. Given we only need the diagonal elements of the Hessian we can write

$$
\lambda_j(i) = \lambda_{j-1}(i) + \sum_{t \in \tau_j} v_{tk}^n (1 - v_{tk}^n) \eta_t(i)^2
\tag{22}
$$

## Results

### Behavioural results

Our main hypothesis is that learning will be facilitated for tasks that share a common subspace where facilitation of learning could be manifested as faster and/or more accurate learning. In order to test the effect of subspace on participants performance, we divided the analysis into two parts. First, we calculated how participants performance in the second task was correlated to that in the first, and then tested whether these correlations differed as a function of subspace. Second, we performed a two-way mixed ANOVA with dependent variable accuracy and independent variables of task (first or second) and subspace (same or different).

**Positive versus negative correlations in same versus different subspace.** Participants performance in Task 2 correlated with their performance in Task 1 when the second task was in the same ($r(38) = 0.42$, $p = 0.007$) but not different ($r(38) = -0.089$, $p = 0.584$) subspace. The two correlations were significantly different from each other (Fisher's z-transform $z = 2.31$, $p = 0.021$).

We then tested whether this effect depended on the Task 1 subspace (Add/Sub) with data and lines of best fit shown in Fig 4. For addition, participants performance in Task 1 significantly correlated with their Task 2 performance for the same ($r(38) = 0.583$, $p = 0.007$) but not different ($r(38) = -0.236$, $p = 0.317$) subspace. These two correlations were significantly different from each other (Fisher's z-transform, $z = 2.647$, $p = 0.008$). The signs of the effects and significant inferences are consistent with the overall picture.

In the subtraction condition, participants performance in Task 1 was not significantly correlated with their Task 2 performance for the same ($r(38) = 0.317$, $p = 0.174$) or different ($r(38) = -0.129$, $p = 0.588$) subspace. These two correlations were not significantly different from each other (Fisher's z-transform, $z = 1.333$, $p = 0.182$). The signs of the effects are consistent with the overall picture but there were no significant inferences.

We do not know why there would be no significant correlations for the Sub subspace but note that the variance of Task 1 accuracies is significantly lower for Sub1 than Add1 (`Std Dev` = 0.06 for Sub, 0.11 for Add, Levene's test $F(1, 78) = 10.94$, $p = 0.001$). Generally, lower variances make it more difficult to detect co-variances/correlations.

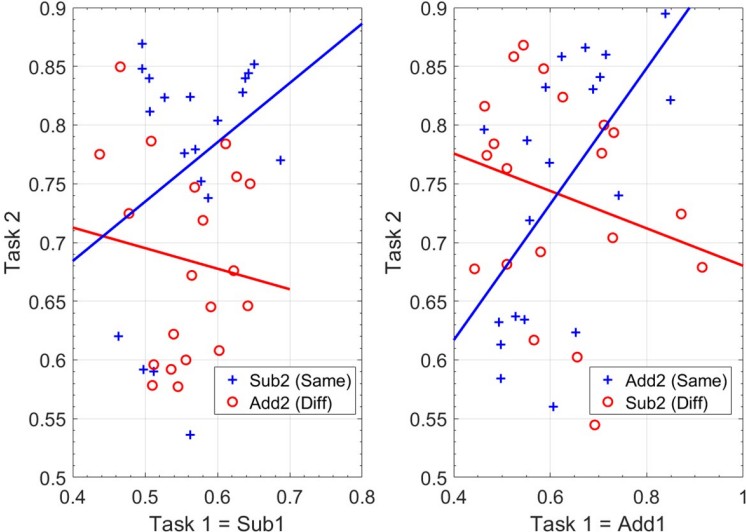

**Fig 4. Correlations over subjects.** For Task 1 = Add1 (right panel), performance in Task 2 is significantly positively correlated with performance in Task 1 when the second task is in the same subspace (blue line and crosses), but negatively when Task 2 is in a different subspace (red line and circles), and the difference in these correlations is significant. For Task 1 = Sub1 (left panel), these effects are not significant although the pattern is similar.

**Increases in Task 2 accuracy for same versus different subspace.** We then performed a two-way mixed design ANOVA with dependent variable accuracy and independent factors of (i) task (1/2) and (ii) subspace (same/different). Overall, we found only a main effect of task (Task, $F(1,78) = 103.71$, $p < 0.001$; Subspace, $F(1,78) = 2.043$, $p = 0.156$; interaction, $F(1,78) = 1.674$, $p = 0.199$). We then performed two separate two-way mixed design ANOVAs for the Add and Sub subspaces. For Add, only the main effect of task was significant (Tasks, $F(1,38) = 29.845$, $p < 0.001$; Subspace, $F(1,38) = 0.024$, $p = 0.876$; interaction, $F(1,38) = 0.001$, $p = 0.974$). For Sub, we found all the main effects and the interaction to be significant (Tasks, $F(1,38) = 100.86$, $p < 0.001$; Subspace, $F(1,38) = 5.669$, $p = 0.022$; interaction, $F(1,38) = 5.195$, $p = 0.028$). The increases in Task 2 accuracy for Same versus Different Subspace were 4.3% overall, 8.2% for Task 1 = Sub1, and 0.4% for Task 1 = Add1. See the Supporting information for plots on the mean accuracies for each combination of Task 1/2.

Overall, the empirical subspace effects are a significant correlation difference for the Add subspace, and a significant interaction (improvement in mean accuracy) for the Sub subspace.

## Modelling results

We used the neural networks described earlier with the following model and optimisation parameters: 4 hidden units per output sub-network, convergence tolerance = 0.001, accuracy threshold pcT = 0.60, MaxIterations = 64, and GELU activation functions in the output sub-networks for Task 1. GELU were preferred over RELU activation functions as, in preliminary work, they produced more similar performance levels on Sub1 and Add1 mappings. For the Task 2 output sub-networks we used linear activations. This was motivated by the fact that these tasks are linear functions of the first layer hidden units and, empirically, this led to better performance on Task 2. We then ran Sequential Bayesian Learning in one of two modes: SBL over tasks, and SBL over blocks (and tasks). The following sections on "Minimal Capacity Network", "Increase Capacity Network" and "Reduced Precision Representation" are based on the SBL over Tasks approach.

**Sequential Bayesian learning over tasks.** First, we created a data set for each mapping as follows. We used 100 input stimuli, $u_t$, drawn from a uniform distribution covering input space. These inputs were presented to a neural net model whose parameters were sampled from their prior distribution (see Eq 10 and the following paragraph). This network then made decisions, $d_t$, by sampling from neural net outputs (see Eq 3—highly stochastic decisions as none of its parameters were yet tuned) and received rewards $r_t$ according to one of the mappings from the behavioural experiment (Sub1, Add1, Sub2, Add2—see section on "Stimulus-Reward Maps"). This created a data set, $R_j = \{u_t, s_t, d_t, r_t\}$ with $t = 1..100$, for that mapping. This was repeated to create a data set for each mapping. The accuracy of a model was then measured using the average probability of being correct (also known as the average trial likelihood—see section on "MAP Estimation") as computed over the training data.

We then tuned the accuracy of Task 1 learning to broadly match the behavioural data, by changing the maximum number of "starts", `maxstarts`, of the multistart optimisation algorithm (see MAP Estimation section). We obtained average task accuracies of 0.56, 0.63, 0.67, 0.69 and 0.73 for `maxstarts` equal to 1, 2, 3, 4 and 8 respectively. This parameter helps the optimiser avoid local maxima by restarting the optimisation with a different initialisation. In what follows we used `maxstarts = 3` and ran 40 simulations per Task 1/2 combination as per the human experiments. Because we were using SBL over tasks, the prior over network parameters for Task 2 was the posterior from Task 1 (see Eq 9). We emphasise that no model parameters were specifically tuned to the particular subspace (Add or Sub) or to individual subject data. The models were simply provided with the above parameter settings, and the same stimuli and reward functions provided to the participants.

**Minimal capacity network.** Here we present results obtained with a *minimal capacity* neural network model having only a single hidden unit in the first layer.

For Task 1 = Sub1, Task 1 performance was 0.66, and Task 2 performance was 0.77 for same and 0.64 for different subspaces. Same-subspace Task 2 accuracies were significantly higher than Task 1 accuracies ($t(39) = 4.76$, $p < 0.001$) and Task 2 accuracies were significantly higher for same versus different subspace ($t(39) = 3.46$, $p = 0.001$). The correlation between Task 2 and Task 1 performance was significantly positive ($r = 0.38$, $p = 0.017$) for same subspace and negative ($r = -0.78$, $p < 0.001$) for different subspace.

For Task 1 = Add1, Task 1 performance was 0.64, and Task 2 performance was 0.76 for same and 0.65 for different subspaces. Same-subspace Task 2 accuracies were significantly higher than Task 1 accuracies ($t(39) = 4.79$, $p < 0.001$) and Task 2 accuracies were significantly higher for same versus different subspace ($t(39) = 2.57$, $p = 0.014$). The correlation between Task 2 and Task 1 performance was significantly positive ($r = 0.47$, $p = 0.002$) for same subspace and negative ($r = -0.68$, $p < 0.001$) for different subspace.

Thus, these modelling results show transfer effects of the sort exhibited in the behavioural data i.e. both increases in Task 2 performance, and correlations between Task 1 and Task 2 performance.

Quantitatively, the standard deviations of hidden unit parameters were 24 times smaller in the posterior (after learning Task 1) than the prior (before Task 1). The figure of 24 is an average over all weights in the hidden unit and over both Add and Sub Tasks. The precisions were thus $24^2 = 576$ times higher in the posterior (after Task 1) than the prior (before Task 1). As the prior (before Task 2) is set to the posterior (after Task 1) this strongly constrains the Task 2 solution to be close to the Task 1 solution (see last term in last row of Eq 18).

**Increased capacity network.** We then repeated the simulations but this time with an *increased capacity* neural network model having two hidden units in the first layer. All transfer effects disappeared.

For Task 1 = Sub1, Task 1 performance was 0.65, and Task 2 performance was 0.79 for same and 0.79 for different subspaces. Same-subspace Task 2 accuracies were significantly higher than Task 1 accuracies ($t(39) = 6.87$, $p < 0.001$) but Task 2 accuracies were not significantly higher for same versus different subspace ($t(39) = -0.023$, $p = 0.982$). Correlations between Task 2 and Task 1 performance were not significant for same ($r = 0.05$, $p = 0.743$) or different ($r = 0.26$, $p = 0.110$) subspace.

For Task 1 = Add1, Task 1 performance was 0.61, and Task 2 performance was 0.78 for same and 0.78 for different subspaces. Same-subspace Task 2 accuracies were significantly higher than Task 1 accuracies ($t(39) = 6.67$, $p < 0.001$) but Task 2 accuracies were not significantly higher for same versus different subspace ($t(39) = 0.118$, $p = 0.907$). Correlations between Task 2 and Task 1 performance were not significant for same ($r = -0.31$, $p = 0.053$) or different ($r = 0.04$, $p = 0.786$) subspace.

These results show that no transfer effects were evident in the increased capacity network, suggesting that a minimal capacity network may be an important factor underlying the behavioural results.

Quantitatively, the standard deviations of hidden unit parameters were 19 times smaller in the posterior (after learning Task 1) than the prior (before Task 1). The figure of 19 is an average over all weights in both hidden units and over both Add and Sub Tasks. The precisions were thus $19^2 = 361$ times higher in the posterior (after Task 1) than the prior (before Task 1). This is a smaller increase than for the minimal capacity network, thus rendering Task 2 solutions somewhat less constrained to be similar to Task 1 solutions (see last row of Eq 18). However, we expect that the main factor in the loss of transfer effects is the increased representational capacity of the network (the required subspace for the Task 2 mapping can be implemented by either hidden unit or distributed over both).

**Reduced precision representation.**   We also repeated the simulations with the minimal capacity network but this time resetting the posterior precision of network parameters from Task 1 to their prior precision at the beginning of learning. All transfer effects disappeared.

For Task 1 = Sub1, Task 1 performance was 0.62, and Task 2 performance was 0.78 for same and 0.77 for different subspaces. Same-subspace Task 2 accuracies were significantly higher than Task 1 accuracies ($t(39) = 6.43$, $p < 0.001$) but Task 2 accuracies were not significantly higher for same versus different subspace ($t(39) = 0.307$, $p = 0.761$). Correlations between Task 2 and Task 1 performance were not significant for same ($r = 0.17$, $p = 0.302$) or different ($r = -0.30$, $p = 0.063$) subspace.

For Task 1 = Add1, Task 1 performance was 0.67, and Task 2 performance was 0.77 for same and 0.77 for different subspaces. Same-subspace Task 2 accuracies were significantly higher than Task 1 accuracies ($t(39) = 3.55$, $p = 0.001$) but Task 2 accuracies were not significantly higher for same versus different subspace ($t(39) = 0.086$, $p = 0.932$). Correlations between Task 2 and Task 1 performance were not significant for same ($r = 0.07$, $p = 0.670$) or different ($r = -0.00$, $p = 0.983$) subspace.

These results show that no transfer effects were evident with reduced precision representations, suggesting that Bayesian estimation may be an important factor underlying the behavioural results.

Reducing the posterior precision effectively removes the protection afforded by Sequential Bayesian Learning to the newly learnt representation, thus allowing it to be overwritten when learning Task 2 (quantitatively, the precision variable, $\lambda_{j-1}$, in the last row of Eq 18 is on average 576 times smaller than for the minimal capacity network—see above section). This results in Task 2 solutions being only very weakly constrained to be similar to Task 1 solutions, thereby eliminating the subspace effect.

**Sequential Bayesian learning over blocks and tasks.** We now report results using Sequential Bayesian Learning over blocks and tasks for the minimal capacity model. We chose our block size to be 25 trials as preliminary analysis (see S1 Text on "Within-versus-Between Block Learning") found there was demonstrable learning within the 50 trial blocks in the empirical data. In SBL over blocks and tasks, the prior over network parameters for learning from data block $j$ is the posterior from block $j − 1$ (see Eq 6). Decisions on data from block $j$ were made by the network before training on that data. The accuracy of a model was assessed using the average probability of being correct (also known as the average trial likelihood—see section on "MAP Estimation"), as computed over the test data set (we refer to this as "test data" as the model has not yet been trained on it). Learning accuracies were then averaged over neighbouring 25-trial blocks to present the model learning curves in Fig 6 (right panel). The equivalent learning curves for the behavioural data are shown in the same Figure (left panel). The empirical data show averages over 80 subjects, 40 in each group (same/different subspace). The simulated data are from a minimal capacity neural net as described above, with 40 simulations per group. The simulated data exhibit similar transfer effects to the behavioural data.

For Task 1 = Sub1, Task 1 performance was 0.62, and Task 2 performance was 0.67 for same and 0.54 for different subspaces. Same-subspace Task 2 accuracies were significantly higher than Task 1 accuracies ($t(39) = 3.21, p = 0.003$) and Task 2 accuracies were significantly higher for same versus different subspace ($t(39) = 3.61, p < 0.001$). The correlation between Task 2 and Task 1 performance was significantly positive ($r = 0.74, p < 0.001$) for same subspace and negative ($r = −0.56, p < 0.001$) for different subspace. These were significantly different from each other (Fisher's Z transform: $p < 0.001, z = 6.77$).

For Task 1 = Add1, Task 1 performance was 0.62, and Task 2 performance was 0.68 for same and 0.54 for different subspaces. Same-subspace Task 2 accuracies were significantly higher than Task 1 accuracies ($t(39) = 2.90, p = 0.006$) and Task 2 accuracies were significantly higher for same versus different subspace ($t(39) = 3.42, p = 0.002$). The correlation between Task 2 and Task 1 performance was significantly positive ($r = 0.81, p < 0.001$) for same subspace and negative ($r = −0.58, p < 0.001$) for different subspace. These were significantly different from each other (Fisher's Z transform: $p < 0.001, z = 7.64$).

The behavioural and neural net data are therefore similarly matched in terms of the positive versus negative correlations for same versus different subspace, and relative increases in Task 2 performance for same versus different subspace. But there are also a number of discrepancies. For example, accuracies at the beginning of the second task experience a sudden drop for the model but not for behaviour, and increases in performance are rather sudden for the model but more gradual for behaviour. These discrepancies are addressed in the Discussion.

## Discussion

We found evidence from our behavioural results in support of our main hypothesis that learning would be facilitated (positive transfer) for tasks that share a common subspace. However, the nature of these transfer effects depended on the subspace. For the Add subspace, transfer effects manifested as positive correlations between Task 2 and Task 1 accuracy (Fig 4, right panel). Whereas for the Sub subspace, transfer effects manifested as higher average accuracy in Task 2 (Fig 5, top right). We do not have an explanation as to why these transfer effects should be different. Clearly, more empirical data is required over a larger number of paired tasks to investigate further.

In our modelling work we found that a minimal capacity neural net model, with a single unit in the first hidden layer, trained using sequential Bayesian learning produced transfer

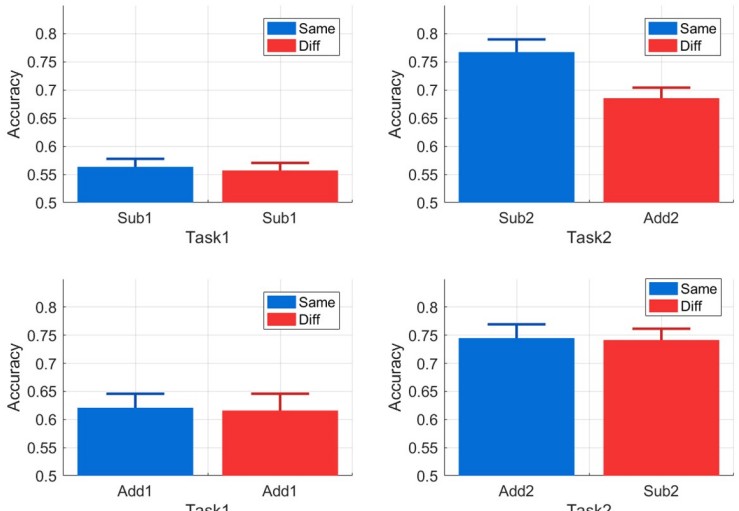

**Fig 5. Increases in Task 2 Accuracy for Same versus Different Subspace.** The barplots show the mean accuracies for Tasks 1 and 2 as a function of whether the second Task is in the same subspace as the first. These results are shown separately for Task 1 = Sub1 (top row) and Task 1 = Add1 (bottom row). For Task 1 = Sub1 there is a significant increase in Task 2 accuracy (of 8.2%) for same versus different subspace (top right). For Task 1 = Add1, mean Task 2 performances are not significantly affected by subspace (bottom right). The error bars indicate the standard error of the mean.

effects that were broadly consistent with the empirical data (Fig 6). This model produced positive correlations between Task 1 and Task 2 accuracy, and differences in Task 2 accuracy, for both subspaces. We then investigated two variants of this approach. First, an increased capacity model with two units in the first hidden layer. Second, a reduced precision model in which sequential Bayesian learning was interfered with by reducing the posterior precision after

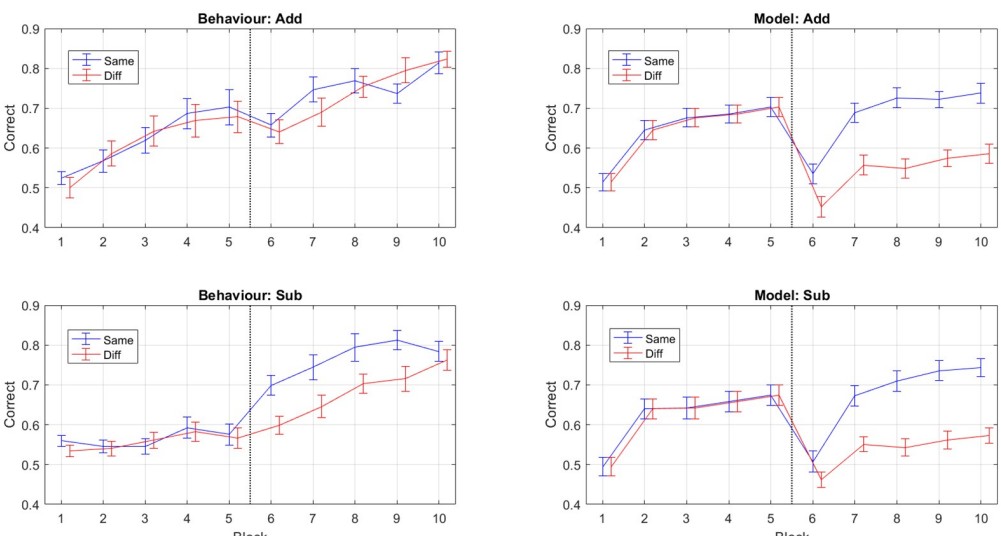

**Fig 6. Behavioural and Model Learning Trajectories.** The behavioural learning trajectories are averaged over 40 subjects for each of the Sub and Add subspaces. The model learning trajectories are 40 simulations from the minimal capacity neural net model for each of the Sub and Add subspaces. Each block comprises 50 trials and the vertical line denotes that blocks 1 to 5 are from Task 1, and 6 to 10 from Task 2. All error bars indicate the standard error of the mean and the red and blue curves have been offset to improve readability.

learning Task 1 to its prior level. All transfer effects disappeared for both of these variants indicating that both a minimal capacity "bottleneck" and sequential Bayesian learning are necessary mechanisms for replicating behavioural findings in this modelling framework.

## Negative transfer effects

In our behavioural experiments participants only performed Task 1 once, and were not told that they would be performing it again. This may have led to an expectation that they would not need to retain Task 1 representations. If this were the case, an increased capacity network would not be needed; a minimal capacity representation would suffice that could be overwritten during the second task. However, we do not know the subject's expectations because they were not explicitly manipulated. This does however motivate a future experiment. If participants were to be told that they will be tested again on Task 1, after learning Task 2, then the prediction is that we will not see negative transfer effects when Task 2 is in a different subspace to Task 1. Such an experiment would speak to a distinction in the neural network literature [8] in which two of the major approaches in the area of transfer/continual learning are to use (i) dynamic architectures in which new representational capacity is added for each new task to be performed and (ii) regularisation approaches in which a fixed architecture is used but regularisation prevents forgetting.

## Future modelling work

There are a number of discrepancies between the model and behavioural learning trajectories. First, accuracies at the beginning of the second task experience a sudden drop for the model but not for the behavioural trajectories. This is likely an artefact of using a modelling approach in which parameters are updated after each block of trials rather than after each trial. This necessarily means that accuracies on the first block of the second Task will be close to chance level.

Second, for the Sub subspace data, same/different trajectories appear to converge towards the end of Task 2 for the behavioural data but not for the model. This might suggest, for example, that there is a recovery mechanism in place in which tight inappropriate priors (inappropriate for different subspace subjects) are replaced by vaguer priors, allowing a more standard learning trajectory to evolve. In preliminary work we had proposed a mechanism in which the prior precision is gradually reduced if learning does not go well. This could be replaced by more formal models, for example with mixture model priors allowing switching from one prior to another during learning. However, as this empirical effect is only evident for one of the experimental groups (Sub not Add) we have decided to postpone further modelling until more data is available.

The model we have presented employs a within-block multistart optimisation procedure such that if estimated parameters do not provide a sufficiently good solution, the estimation is repeated, with a total of `maxstarts = 3` model fits allowed per block. This serial model fitting process is biologically implausible but could potentially be implemented using parallel architecture and may fit in with evidence that up to three or four decision making strategies can be simultaneously updated and monitored [29]. One possibility for future modelling, however, would be to use a moving window of samples to which the model is fitted, rather than splitting the samples into non-overlapping blocks. This would remove the "blockiness" of the results referred to above, and the inherent stochasticity of the approach may remove the need for multistart optimisation. Such an approach has been used to good effect in recent work on dynamical models [30].

## Elemental and configural learning

One distinction in the category learning literature is between elemental (or linear feature-based) learning and configural (or object-based) learning. Duncan et al. [18] showed that people switch between learning styles as a function of the empirical contingencies in the data (e.g. elemental if reward functions are indeed a linear function of features). They also noted, however, that a proportion of participants persisted with a configural strategy even when a more efficient elemental strategy would have sufficed.

In a similar experiment, Farashahi et al. [31] showed that people shift from elemental to configural representations as they learn. They describe an elemental RL agent in which values are learnt for each discrete setting (out of $M$ settings) of each input variable (out of $D$ inputs)—thus requiring up to $D \times M$ values to be learnt. The overall value of a stimulus is then given by a linear combination of feature values. This is to be contrasted with an "object-based" RL agent which learns a value for each object (or "configuration"). Given that there are $M^D$ possible objects, this requires learning a potentially much larger set of parameters. Empirical results demonstrated that people initially employed a (linear) feature based strategy and later switched to an object-based one. This took place even when the true contingencies were not linear.

One of the goals of the current paper was to explore mechanisms underlying learning of rather general nonlinear mappings (which are in turn composed of an input to latent space function (subspace) and a latent space to output function). Both elemental and configural learning approaches are, however, highly suboptimal for these tasks, the elemental strategy because it is linear, and the configural strategy because it is statistically inefficient (having a number of parameters that grows exponentially). More specifically, an elemental learning strategy could be applied for the Sub2 and Add2 linear value mappings in the current study, but would be unsuccessful for the Sub1 and Add1 reward functions which are nonlinear. A configural strategy would be highly inefficient for any of the tasks due to the large number of configurations, $M^D = 25$.

## Learning accuracy across mappings

There is a long-standing debate in the category learning literature about whether and how humans can learn non-linearly separable categories [32–34]. Medin et al. [33] and Levering et al. [34] both find that non-linearly separable categories are easier to learn than linearly separable ones. Their experiments used three binary input features (resulting in only 8 unique stimulus vectors, 6 of which were shown during learning) and binary classification labels. Importantly, the input vectors were chosen so that the "well-formedness" of the categories (and therefore, presumably, the maximum achievable classification rates) were matched across linear and nonlinear tasks.

For the tasks in the current paper, the maximum achievable classification rates were closely matched across the linear (93 per cent) and nonlinear (95 per cent) tasks. In contrast to previous work, we found that the linear mappings were easier to learn than the non-linear mappings (mean accuracy = 73.5% for linear, 58.9% for nonlinear, see Supporting information for further details). However, as the linear tasks were always performed after the nonlinear tasks, this could be due to an order effect, or indeed the transfer effects that are the main interest of this paper. The linear/nonlinear issue could be addressed in a future experiment in which participants learn just a single task.

There is a literature on "human function learning" that presents participants with data points sampled from one-dimensional functions and asks them to predict where future samples will be drawn from [35, 36]. This literature shows that people have a preference for linearly increasing rather than decreasing completions i.e. positive rather than negative functions.

These findings on 1-dimensional functions are perhaps difficult to extrapolate to the 2-dimensional functions used in the current paper. Empirically, we did not find that (collapsing across task 1 and 2) the add subspace functions were learnt more accurately than the subtract subspace functions (see S1 Text for details). One difference we did find was that more participants correctly declared the rule underlying the Sub2 map than the Add2 map (see S1 Text).

## Rule-based learning

Subjects who performed the Sub1 and Sub2 or Add1 and Add2 tasks did better (than those who performed Sub1 and Add2 or Add1 and Sub2), but was this really because the tasks were in the "same subspace". Are there not other similarities among these tasks? For example, that both required the same logical operation or rule-based operation as an intermediate step? This speaks to a body of work in rule-based learning. One approach to this topic is the "Rational-Rules (RR)" model [37] which formalizes a statistical learner that operates over the space of Boolean propositional logic expressions e.g. "A or B", "A and B", "A or (B and C)". In an fMRI study, Ballard et al. [38] found that the pattern of striatal responses was more consistent with prediction errors derived from such a rule-learning model than a Reinforcement Learning model. We accept therefore that there could be an ambiguity in interpretation here and that resolution of this issue requires further empirical work, perhaps with experiments using nonlinear and/or multivariate subspaces that are not readily expressible using rational rules.

## Declarative learning

In additional statistical analysis presented in S1 Text we show that subjects who were able to declare a correct rule-based strategy also showed a stronger subspace effect. We also show, however, that subjects who performed better in the first task also showed a stronger subspace effect. Further analysis then showed these two effects to be moderately collinear (as those who declared a correct rule-based strategy also performed better on the first task). Therefore, with the current data, we are unable to infer which of these factors (declarative learning or accurate learning) drives the subspace effect. Again, further experiments are required perhaps using nonlinear and/or multivariate subspaces.

## Creation or selection of representations?

Are new representations created i.e. features learnt? Or, are pre-existing representations prioritized as potentially useful and selected from, as proposed by Collins and Koechlin [29]. For example, there may be representations in brain regions encoding for numerosity [39] that already encode differences and sums over numbers of items. An additional component in the model proposed in [29] is a process that creates new stimulus-response mappings from old ones. It could be that the offline learning algorithm we have described in this paper, or some similar process, plays this creative role.

## Structure learning

This paper fits in more broadly with previous studies of structure learning which show that people take advantage of shared structure across tasks. For example, Costa et al. [40] studied rhesus monkeys taking part in a probabilistic two-armed bandit reversal learning task in which monkeys were exposed to a distribution of reversal times and were able to make use of this information during decision making. Tomov and Gershman [41] studied people engaged in a novel two-step decision making task, finding evidence that human subjects use a multitask learning strategy that maps previously learned policies to novel scenarios. As with our paper,

the rewards were a function of multiple input features and this changed across tasks. In theoretical and simulation work, Franklin and Frank [42] address the problem of transfer learning in a Markov Decision Process context by designing a non-parametric Bayesian agent that can generalise across state-transition functions, reward functions or both.

Radulescu et al. [43] review recent research which suggests that, in complex learning tasks, human behaviour is consistent with an integrative model in which approximate Bayesian inference acts as a source of selective attention, allowing Reinforcement Learning (RL) to focus on the relevant dimensions for decision making. Within the Bayesian approaches, Latent Causal Models (also known as Non-Parametric models) organise experience into similar episodes, and Probabilistic Programming allows rules based on logical operations to be inferred. In principle, it may be possible to adapt the Latent Causal Model framework to the study of transfer learning, for example, by allowing for common causes among tasks but adapting contingencies between causes and outputs. This is an avenue we will explore in future work.

## Transfer learning

The study of transfer learning has a long history in psychology [13], and more recently in the fields of cognitive training and cognitive neuroscience [44]. A key qualitative concept here is the notion of near versus far transfer where distance reflects how similar the different learning contexts are. This may naturally map onto the quantitative measures defined in Bayesian learning e.g. the probability density of task-two feature parameters under the task-one posterior. Noack et al. [45] propose a theory-driven approach to studying transfer effects in cognitive training research. They argue that data should be analysed within the context of theoretically motivated (using hierarchical cognitive process models) and/or latent factor analysis methods, so that inferences can be made at the level of latent processes. The work in this paper concurs with this latent and hierarchical perspective, but whereas Noack et al. deconstruct existing batteries of cognitive tasks, our goal is to design new tasks with better defined relationships among latent and observed variables.

Building on long established models of cognitive control, Musslick and Cohen [46] present a three-layer neural network architecture with stimulus layer, hidden layer and output layer but augmented with task units that affect the hidden and output layers. Learning in these networks allows a mapping between task and hidden units such that irrelevant hidden units are inhibited. The network is trained on multiple tasks with simulations showing interference between tasks that required activation of common hidden units (representations). By adding temporal persistence to the hidden and output layer activations (reflecting the dynamics observed in biological networks) they were able to explain well-established phenomena such as the psychological refractory period. This important issue of task switching and maintenance has been neglected in our paper. We have instead assumed that only the relevant output subnetwork is engaged while the other is inhibited, without providing a mechanism for this.

Flesh et al. [47] compared human learners and neural net learners in transfer learning tasks involving categorisation of naturalistic images of trees. As expected, the neural network suffered from catastrophic forgetting when samples of each task were blocked rather than interleaved. Conversely, human performance was better if the samples were blocked rather than interleaved. They showed that neural net performance on blocked data could be improved by pre-training using a generative model approach. This was implemented using an autoencoder in which a two-dimensional "bottleneck" layer enabled learning of the appropriate two-dimensional subspace. This subspace comprised the two relevant features that predicted reward across tasks and is analogous to the one-dimensional subspaces studied our paper.

Wu et al. [48] studied the transfer of knowledge between spatial and conceptual domains. They specified a series of two-dimensional reward maps which were identical over both domains and found transfer effects from spatial to conceptual domains but not vice-versa. Impressively, transfer was examined using eighty different reward maps (rather than the four examined in the current paper). Subject's behaviour was well described by Gaussian Process (GP) models (as in [19]). GPs are an ideal choice for the goals of their study but do not break down mappings compositionally as in the current paper (such that mappings can share a subspace but have different subspace to reward functions).

Wang et al. [49] present simulations of a meta-reinforcement learning agent in which a recurrent neural network, posited to reside in prefrontal cortex, has adjustable parameters that are trained using RL, not on a single task, but instead in a dynamic environment comprising a series of related tasks. The activation dynamics of this network then manifest a second within-task RL algorithm that is automatically tuned to the task at hand. The model explains a broad variety of well-established phenomena including an updated version Harlow's original learning to learn (multitask learning) paradigm in which, after a series of learning episodes, monkeys (and the Meta-RL agent) exhibit single-shot learning.

Yang et al. [50] also present simulations of a recurrent neural net model of frontal cortex showing how it can learn twenty different cognitive tasks. Interestingly their model employs a regularisation approach, similar to the EWC method used in this paper, to prevent parameters of 'older' tasks being overwritten during learning. They also analyse the representations formed noting that transfer can be mediated either by clustering of parameters over tasks or by the development of compositional representations (of the sort investigated in the current paper).

## Machine learning

The role of task units examined in Musslick and Cohen [46], in which task units can inhibit hidden units, has been examined as a potential mechanism for aiding multitask learning by Masse et al. [51]. Their studies, using high-dimensional pattern recognition problems, also examined an alternative "gating" strategy in which task units could directly "gate" hidden unit activations (thus mimicking neuromodulation in the brain), such that a proportion of hidden units are gated (set to zero) for any given task. Both of these proposals were examined in combination with EWC [11] with the findings being that the gating strategy produced better empirical results.

The starting point of our paper was to leverage recent conceptual and algorithmic progress in machine learning to define new experimental psychology tasks and computational models, with the longer term goal of better understanding human multitask and transfer learning. To do this we made use of a sequential Bayesian regularization approach to prevent catastrophic forgetting. This literature, however, is rich with other quantitative ideas about how to define relationships among tasks which could inform the design of future experiments. These include, for example, "sluice" and "cross-stitch" networks [5] which automatically infer how to share subspaces at multiple hierarchical levels and across multiple tasks. Sequential Bayesian learning for neural networks is also being applied to the more challenging problem of continual learning and is producing state-of-the-art performance on benchmark problems [52].

## Supporting information

**S1 Text. Supporting information contains the derivations of the neural network model and further analyses of the behavioural data.**
(PDF)

## Acknowledgments

We thank Francesco Silvestrin for piloting the experiment and John Spencer and Tom Sambrook for feedback on earlier experimental designs.

## Author Contributions

**Conceptualization:** Nicholas Menghi, Will Penny.

**Data curation:** Kemal Kacar.

**Formal analysis:** Nicholas Menghi, Will Penny.

**Investigation:** Nicholas Menghi.

**Methodology:** Nicholas Menghi.

**Supervision:** Will Penny.

**Writing – original draft:** Nicholas Menghi, Will Penny.

**Writing – review & editing:** Will Penny.

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
