## [Decision Letter · Decision Letter 0]

8 Sep 2020

Dear Dr Menghi,

Thank you very much for submitting your manuscript "Multitask Learning over Shared Subspaces" for consideration at PLOS Computational Biology.

As with all papers reviewed by the journal, your manuscript was reviewed by members of the editorial board and by several independent reviewers. In light of the reviews (below this email), we would like to invite the resubmission of a significantly-revised version that takes into account the reviewers' comments.

We cannot make any decision about publication until we have seen the revised manuscript and your response to the reviewers' comments. Your revised manuscript is also likely to be sent to reviewers for further evaluation.

Sincerely,

Samuel J. Gershman

Deputy Editor

PLOS Computational Biology

Editor's comments:

The reviewers cover most of the key issues I noticed in the paper. The one additional point I want to raise is that some of the statistical results are pretty weak (even with the liberal use of one-sided tests, some of the results are not very strong). Weak results by themselves are not problematic but in this case there is the risk of false positives and p-hacking, where a two-sided test is sometimes done first (revealing no significant effect), followed by a one-sided test (revealing a significant effect).

Reviewer's Responses to Questions

**Comments to the Authors:**

Reviewer #1: In this paper, the authors show that humans can perform a new task better, at least initially, if the relevant features are the same as those from a previous task. Using computational modeling, they show that this effect be accounted for by a form of “soft parameter sharing“ (SPS): a technique employed in multitask learning, whereby the parameters for the new task are initialized with parameters from a previous task.

In particular, on each trial, the inputs u (u1 = # slices on pie 1, u2 = # slices on pie 2) are mapped onto latent features x by a linear transformation, x = Au. The features are then mapped onto outputs in potentially nonlinear ways. Two tasks are said to share the same subspace if the optimal A is the same (even if the mapping from features to outputs is different). The authors explore three computational alternatives for the way in which humans learn to perform the tasks:

- soft parameter sharing (SPS): A for task 2 is initialized with A from task 1, but is potentially overridden if performance is not good

- Hard parameter sharing (HPS): A for task 2 = A for task 1

- RBF: A = I for both tasks, i.e. x = u

They show that the behavior of SPS is most consistent with the human data.

Overall, I found the paper to be relatively well-written and clear. It is covering a very timely question: the ability of humans to learn new tasks so quickly is thought to rely on inductive biases from past experience, but the details of how this process unfolds and how it might be implemented in the brain are far from clear. Multitask learning provides a formal framework for acquiring such biases and modifying them with experience, and thus can naturally serve as a source of hypotheses of how the brain might be doing it. While on the surface, the result might not seem so surprising — if you give people a similar task, of course they do better — formalizing what “similar” means (same latent features, as in this paper, as opposed to e.g. same output mapping) and studying the nuances of how exactly they “do better” (e.g. better at first vs. throughout) are essential steps in testing hypotheses about the underlying computational process. Thus I think this paper is an important step towards elucidating the details of multitask learning in humans.

I do have a few suggestions on what the author can do to improve it.

First, I think the authors should include a No Parameter Sharing (NPS) model, which is the same as SPS, except the A’s are not constrained to be similar across task. This seems like a plausible alternative, which will do better initially in the different subspace condition compared to SPS. Naturally, it will not have an advantage in the same-subspace condition, so ultimately it is a bit of a strawman model, however I think it is more plausible than the RBF model and I suspect some subjects might be relying on something like that (i.e., not transferring any knowledge and learning the new task from scratch).

Second, I would consider adding a version like SPS but without adjusting the precision as in Eq 15. That is, just continual learning, as if you’re still getting trained on the same task even though you’re on task 2. Note that this is not the same as HPS, since A can still keep changing for task 2 (whereas for HPS it is fixed), it’s just that the rate at which it is changing is fixed. I think this simpler form of learning (which in fact might not even be considered multitask learning, as from the point of view of the agent, there’s just a single task) might account for the data as well as SPS.

Third, I think the authors could do a better job of visualizing the links between human and model behavior. I recommend reorganizing the figures (and perhaps the exposition), such that each figure shows the same effect for humans, SPS, HPS, and NPS side-by-side, in the same way and with the same analysis. This would also consolidate the figures a little bit.

Minor:

The notation is not very clear, and I recommend the authors explicitly define all the variables after each equation unless they have already been defined. E.g., equation 2: y, u, mu, etc are not defined. Or equation 3: what is phi? What is n? Some of these are defined only later or not at all. Also, the subscript notation is ambiguous: at first, u_t denotes trial, but then u_1 denotes the first component of u (line 113).

Line 121: did they learn a different task on the different day?

I would add subsections to the Value Network section explaining precisely how SPS, HPS, NPS, and RBF differ from the generic Eq 3, with separate equations. Right now, this information is interspersed throughout.

The section on off-policy learning is confusing: isn’t the same agent learning and also performing the task?

The likelihood function P(R|A) in Eq 6 is not defined anywhere. Also, the notation is a bit ambiguous, as m is already used to denote the RBF centers.

What are the error bars in figure 5 and 7?

What is adjusted data in figure 6?

Line 334 and 337: F statistic and df’s

Figure 8 and the corresponding text seem redundant with figure 6; just a different way of visualizing the same result

Figure 9 seems to be contradicting figure 7 – this itself is an interesting result which should be further looked into and perhaps investigated using computational modeling.

Line 408: what drives the difference between good and bad learners? Is it just the stochasticity in the parameter optimization process? Or the initialization? Or the training?

Relatedly, on line 413, what drives this effect? Is it that the better learners learned the correct feature mapping, or are they just better at learning overall? The benefit of having a computational model is that you can easily answer those questions, whereas it is difficult to answer them for the participants directly.

Relatedly, line 421 – This differs from the human data and bears further discussion, also investigating in more detail what in the computational model drives this difference.

Relatedly, line 436 – what is driving this difference?

Line 469 and the following paragraph – Not sure I get distinction between features and subspaces, as defined here. Any subspace can be defined by a basis set of features, so the two seem isomorphic. I suppose what the authors are getting at is that when the dimension of the subspace is 1 (and hence it can be described by a single feature), there might be simpler strategies that subjects could be relying on?

Reviewer #2: Menghi et al. present results from a behavioral task that assesses humans’ ability to exploit shared structure between different reward environments (“tasks”). They also develop a computational model of how an agent may exploit this shared structure based on the ML framework of Multitask Learning.

The paper claims that:

* The task is a novel way to study transfer learning

* The model somewhat matches how humans learn

* Shared subspaces are a useful framework for thinking about multitask learning

Overall I am enthusiastic about the approach, but found that the paper could benefit from (1) More focused claims, and relating the findings to several strands in the literature that have already examined transfer learning (albeit under slightly different names); (2) Deeper analysis of ways in which the model succeeds or fails in capturing various aspects of how humans behave in this task.

Major comments:

(1) It is stated only later, but the authors hypothesized that learning is easier for tasks that share a common subspace. This finding on its own is not particularly surprising. As the authors point out in the intro, we know from previous studies of structure learning that people take advantage of shared structure across tasks (e.g. https://www.jneurosci.org/content/35/6/2407, https://www.biorxiv.org/content/10.1101/815332v1 and https://journals.plos.org/ploscompbiol/article?id=10.1371/journal.pcbi.1006116).

That said, the authors have an interesting design which could be leveraged to ask specific questions about how humans actually manage to do this. For example, what is going on with the obvious difference in accuracy between Task 1 and Task 2? Is the main difference that Task 1 requires learning a non-linearity? If so, that is interesting! There is a long-standing debate in the category learning literature about whether and how humans can learn non-separable (NS) categories (see https://psycnet.apa.org/record/2011-17802-001 and https://link.springer.com/article/10.3758/s13421-019-00972-y). One idea is that humans shift from elemental (feature-based) to configural (object-based) representations as they learn (see https://www.cell.com/trends/cognitive-sciences/fulltext/S1364-6613(19)30036-1 and https://www.nature.com/articles/s41467-017-01874-w). Do the authors see evidence of this in their data? Does the model they present display such a shift in representation across learning blocks? How does linear separability interact with humans’ ability to exploit shared structure? And to what extent does learning the NS task first affect parameter sharing?

To be clear, I am not necessarily suggesting that authors answer all these questions in the paper, just pointing out a direction they could take to see how their model can speak to some of the open questions in the literature.

(2) Second, it really wasn’t clear to me just how well the model captures key aspects of human data.

In Figure 7, we see that subjects exploit structure, which manifests as a performance gain early during Task 2 and gradually increases over time. This pattern suggests that people learned some kind of higher order rule, such as the one authors suggested at the bottom of page 4. But in Figures 10 and 11, which show us what the model is doing, it looks like in both conditions the model starts Task 2 much closer to chance and the improvement is then sudden and stays constant. Both of these patterns would be picked up by an ANOVA, but they suggest qualitatively different learning dynamics. So what is going on here? In terms of presentation, it would really help to add the human plot next to Figures 10 and 11 to make the comparison clear.

Similarly, even in the different subspaces condition, it looks like the model starts much lower than humans do (45% performance in Fig. 11 vs. 65% performance in Fig. 7).

It’s striking that Hard Parameter Sharing fails so spectacularly when tasks do not share a subspace. SPS seems critical but maybe even too good compared to humans? Is it possible to vary the degree of “sharedness”, and if so, what would the optimal level be for this task?

It could be that some of these differences stem from reporting Accuracy for humans and Average Rewards for the model, but it’s not possible to know without a side-by-side comparison and directly simulating actions by the trained model.

(3) Were subjects explicitly instructed that the second task was different? Or was the difference only in the framing (sun/rain vs. heads/tails)? This seems important, as the network is strongly encouraged to separate by allowing it knowledge of the task label.

(4) Finally, the authors might want to check out and connect to the work of several authors which is directly relevant for what they are trying to do here:

Wang et al.: https://www.nature.com/articles/s41593-018-0147-8

Musslick and Cohen: https://cogsci.mindmodeling.org/2019/papers/0161/index.html

Minor comments:

I found the training procedure section unclear: how does the NN training interact with sequential Bayes? Do you solve for the MAP and then use that as a training objective? And how does this interact with the log likelihood of *reward*?

Can participants verbally articulate learning some of the rules? That datapoint might be another useful one in determining what is actually being learned

The Fig. 2 caption brought a lot of clarity regarding how subspaces were operationalized. Consider adding some version of this text to the “S-R mapping section” as a preamble. E.g. “Subspaces were operationalized by defining a common feature that, when represented, reduced the task to a rule. In one case this was the sum, in the other the subtraction… “ — and only then present the equation

Please take a pass through the paper and clarify notation as much as possible. Some examples:

* In equation 2, what are u, mu and W? These are only introduced later...

* In equation 3, v and h are not displayed in Fig. 3

Table 1 is helpful in clarifying the design; consider adding columns specifying: the number of subjects, the type of task (weather or coin) and the break

Stimulus-Reward Maps or “Value Functions”?

Correct terminology would be “Reward Function”, as it defines the reward the participant would get in a particular state contingent on action

Figures 10 and 11 have the wrong x-ticks (should be 1:10)

Reviewer #3: REVIEW OF “MULTITASK LEARNING OVER SHARED SUBSPACES”

Summary

This study presents an experiment on transfer between tasks with a

shared or non-shared feature subspace, as well as a model with “soft

parameter sharing” that can flexibly transfer a previously learnt

representation or not. The model is compared with a version which forces

“hard parameter sharing” (always transferring a representation) and

another model (RBF) which does not transfer. The experiment shows a

benefit in performance when tasks share a subspace, and the modeling

results show that only the soft parameter-sharing model shows this

benefit.

Evaluation

The manuscript is generally well-written and clear and the authors

clearly understand the models they developed (and how to make them

work). I find the results of the experiment interesting. Looking at

Figure 2, it doesn’t seem obvious why sharing an additive or subtractive

way to combine the cues would provide a benefit, given the different

ways in which this combined feature is mapped to value. Aspects of the

model are interesting too (e.g., combining Bayesian inference with

neural network models). I didn’t find the comparison between the models

that interesting though. Simulating models that either force complete

transfer or no transfer at all will obviously give the result of

(failed) transfer vs no transfer. As the models are not fit to

participants’ behaviour, the potentially interesting question whether

soft parameter sharing also best describes people in the same subspace

conditions remains unanswered. The RBF model with no transfer further

differs in many respect to the other two models and hence a clear

comparison is difficult. It seems easy enough to implement a model of

the same form as the other two with no transfer (i.e., simply resetting

the prior at task 2 to the same one at task 1), and I wonder why the

authors didn’t include this version. I also feel the data are a little

“over-analysed”, whilst important things are missing. The discussion

lists a number of open questions and limitations, which is honest, but

does leave the reader wondering what can be inferred from the results.

With these things in mind, on balance, I felt a little underwhelmed with

the paper. There are good things there, and some of the issues listed

below can be straightforwardly addressed, but I think the paper would

gain a lot more from running an additional experiment with multivariate

subspaces.

Major issues

1. Apart from the form, no mention is made of open science practices

in the actual manuscript. I strongly urge the

authors to make their data, materials, and analyses scripts

available.

2. The main analysis showing the benefit of shared subspaces consists

of two separate t-tests. However, showing that one test is not

significant, and another is, is not itself a good test or indication

of a significant interaction effect, which is really what the

hypothesis is about (a larger increase in performance for shared vs

non-shared subspaces). the authors should minimally show that this

interaction is directly significant.

3. Is collapsing over subspaces warranted? All analyses collapse over

the subspace type (additive vs subtractive), which makes sense given

the goals of the study, but not so much given what we know about

human function learning (which shows a clear benefit of positive

functions (additive) vs negative functions (the subtractive subspace

can be seen as a combination of a positive and negative function).

The authors should at least show the performance separately for the

additive and subtractive subspaces (and the second task also as a

function of whether the subspace was additive or subtractive in the

first task). I think this may highlight important deviations between

human behaviour and the models (which won’t inherently care whether

the subspace is additive or subtractive).

4. There is a proliferation of analyses, which can lead to increases

false positives. I think the main important analyses can be

conducted with a single (larger) factorial ANOVA model, with task (1

or 2), subspace task 1 (additve, subtractive), subspace similarity

task 2 (same, different), and block as factors. The main effect of

interest would be the “task” by “subspace similarity task 2”

interaction, but the analysis will control for a number of other

factors, which might be of interest as well.

5. I didn’t find the analyses involving a median split on good vs bad

performers that informative. Given the bounded scale of the

dependent variable, it is not so strange that those who initially do

well don’t improve as much as those who initially perform poorly.

6. I imagine the use of online batch learning for the two parameter

sharing models was used in order to allow for the model to flexibly

choose a “right” level of transfer for the soft sharing version.

That, as well as the “check solution is OK or increase the variance

of the prior” seems like a bit of a “hacky” solution to me. At least

as a model of human behaviour, it would predict no performance

increase within blocks, which seems unlikely to me. So that begs the

question whether (apart from being able to transfer or not depending

on the task similarity), the soft parameter sharing model is really

a plausible model of human learning. The way soft parameter sharing

works seems rather reminiscent of things like Dirichlet processes

and Anderson’s model of rational category learning, where a new

representation for a task would be learned whenever it doesn’t fit

an already learned representation very well. If a mixture approach

is taken, where the new task is either from the posterior

distribution of the previous task, or drawn from a more general

prior suitable for new tasks, I think a more principled approach

could be taken, where the posterior over mixture components defines

the weight between transfer and no-transfer. This would have the

benefit of being both “properly Bayesian” and allowing for an fully

online model.

7. The comparison between the two parameter sharing models and the RBF

seems rather unfair. The first two are trained in an offline batch

manner, optimizing the learning rates for each batch, while the last

is trained in an online manner, with a fixed (arbitrary?) learning

rate. Given the many differences, any comparison in model

performance is difficult. If the interest is in transfer vs no

transfer, why not stick to the same general model as the parameter

sharing models (using a completely “refreshed” prior for the second

task). If the interest is in a difference in representation, why not

allow the RBF model to use soft and hard parameter sharing?

Personally, I’d like to see the results of all six models, and all

trained in the same way (preferably online). At the moment, no clear

conclusions can be drawn from comparing RBF to the other two models.

Minor issues

- line 128. What was the duration of break between blocks supposed to

achieve?

- line 364: “overtime” -> “over time”

- Appendix A.3.2 and A.3.3 I don’t think the ANOVA on the models with

“good” vs “bad” learner is overly meaningful. Especially for the

RBF, which uses the same parameters throughout, good vs bad may

depend somewhat on the input, but I doubt there is a meaningful

distinction. For the other two models, which depend on some

randomness in priors, the distinction may be larger (hence the

differences found), but then may be due to rather uninteresting

differences, rather than anything meaningful.

- Appendix B.3 “We therefore infer these effects to be collinear and,

with the present data, we cannot tell which of these factors is

driving the subspace effect.” There are agreed upon measures for

collinearity (which depend on relations between the independent

variables/predictors), such as the tolerance or variance inflation

factor. The finding that effects or not significant if both

predictors are included in a model is likely due to collinearity,

but it is not a suitable test of it.

**Have all data underlying the figures and results presented in the manuscript been provided?**

Reviewer #1: None

Reviewer #2: Yes

Reviewer #3: **No: **The form states data is available, but no details are given.

PLOS authors have the option to publish the peer review history of their article (what does this mean?). If published, this will include your full peer review and any attached files.

Reviewer #1: **Yes: **Momchil Tomov

Reviewer #2: No

Reviewer #3: No
---

## [Decision Letter · Decision Letter 1]

29 Jan 2021

Dear Dr Menghi,

Thank you very much for submitting your manuscript "Multitask Learning over Shared Subspaces" for consideration at PLOS Computational Biology. As with all papers reviewed by the journal, your manuscript was reviewed by members of the editorial board and by several independent reviewers. The reviewers appreciated the attention to an important topic. Based on the reviews, we are likely to accept this manuscript for publication, providing that you modify the manuscript according to the review recommendations.

Sincerely,

Samuel J. Gershman

Deputy Editor

PLOS Computational Biology

[LINK]

Reviewer's Responses to Questions

**Comments to the Authors:**

Reviewer #2: I thank the authors for the thorough revision!

Regarding connections with the literature:

Thanks for the discussion of:

* Structure learning

* Elemental vs. configural learning

* Multitasking NNs

I think it will be very helpful for people interested in the issue of transfer learning to be able to connect with this literature.

If I may suggest moving these bits from the Intro to the Discussion under a larger "Related Work" section, in which authors highlight after each one how their approach advances or relates to each line of work. E.g, move the short para on page 7 to the Discussion, and say that the author's approach is one way to resolve the issue of elemental learning being insufficient for nonlinear functions, and the configural learning being, indeed, very costly. That way the focus will stay on the authors' main contribution. It felt like this comment (creating an explicit Related Work section) may apply to other parts of the Discussion as well.

I also appreciated the discussion of how the issue of linear/non-linear mappings interact with the results, and agree that it is hard to disentangle from an order effect with the current design.

Regarding results:

The behavioral results are much easier to parse, thank you!

In the Discussion when summarizing results, authors may want to note that the performance boost by a shared subspace is task-dependent, in that it holds for Addition, but not Subtraction. You can leave it to future work as to why that may be the case. But it seems important to be precise here, especially because it's a result that points to possible changes to the model.

I'm also now realizing after seeing the new plots that the correlation in performance between Task 1 and Task 2 happens only for Add (Figure 4, page 15, paras 1 and 2 -- this result is not shown in the figure, but mentioned in the text). This is perhaps inconsistent with the result in Figure 6, in which the behavioral boost attributed to the shared subspace happens for Sub, but not Add. What might be the explanation there?

In the learning curves, perhaps use a vertical line to mark the start of Task 2 between Blocks 5 and 6?

Finally, for the section titled "Discrepancies between model and behavioral data": authors may want to rename it to something more positive like, "Future Modelling Work".

Reviewer #3: Review of “Multitask Learning over Shared Subspaces”

Summary

This study presents an experiment on transfer between tasks with a

shared or non-shared feature subspace, as well as three artifical neural

network models with Sequential Bayesian Learning. The experiment shows a

benefit in performance when tasks share a subspace, and the modeling

results show that a “minimal capacity” ANN with SBL matches human

performance better than an “increased capacity” ANN and amodel with

“reduced precision” (which increases the variance of the prior in the

transfer task).

Evaluation

In response to the previous review round, the authors did an extensive

revision of the manuscripts and particularly the modelling. The

statistical analysis of the behavioural results is much cleaner now. The

new modelling is not really comparable to that of the previous

submission, but I like that all models presented derive from a single

framework. However, I find the difference between the restricted and

increased capacity model less intuitive as a manipulation of learning

transfer. Perhaps more importantly, the restricted capacity model with

SBL matches the transfer patterns not by offering benefit for a shared

subspace task, but rather a detriment for a non-shared subspace task. As

an account of learning transfer, that seems rather disappointing.

Major issues

1. The analysis of the behavioural results is much tighter now. But the

results of the two mixed ANOVAs are difficult to interpret without

seeing the data separately for the different types (additive vs

subtractive). The plots (Fig 4 and 5) should really reflect the new

analyses, and not show the data aggregated over the types.

2. The difference between the restricted and increased capacity model

can be more clearly described. What is the effect of setting the

number of hidden units in the first hidden layer to 1 vs 2 on the

solutions that can be achieved? As the results show, it seems

crucial to replicating worse performance in the different subspace

conditions that there is one hidden unit in the first layer, but as

someone who doesn’t work with ANNs often, I find it difficult to

interpret the meaning of this, especially since the reduced and

increased ANNs both use SBL. Is there perhaps a substantial

difference in the precision of the posterior distribution over the

connection weights, such that the increased capacity model can more

easily overcome a “wrong” prior? Or is something else going on?

3. I’m not convinced the new models match the behavioural patterns that

well. The “minimal capacity” model shows not so much a benefit of

same subspace, but a detriment of different subspace. The behaviour

of participants, on the other hand, seems to indicate a benefit of

shared subspace (especially in the Sub condition, where performance

is immediately better in Task 2 than Task 1). Model performance in

Task 2 never exceeds that of Task 1. While the differences between

the model and human behaviour are acknowledged in the Discussion, I

found the explanations not that convincing. In particular, if

participants find the linearly separable tasks easier, that seems to

go against a transfer of a learned “shared subspace” representation.

4. Model performance was matched to behaviour by changing the maximum

number of starts of the optimization routine. It is not immediately

obvious to me where this applies, but I guess its the number of

reductions of the step size in the fminbnd.m routine? Setting this

number at a low value makes it likely that the result is a local,

rather than global, maximum. Why is this chosen to effectively

reduce model performance, rather than other possibilities

(e.g. introducing randomness in the responses, reducing the

effective learning rate by increasing the precision of the prior,

etc). Compared to these latter possibilities, the “maxstart”

parameter seems a rather “hacky” choice. This is of course

subjective, but this particular setting of a numerical optimization

routine seems quite far removed from anything that might be

plausibly differ between participants. Can more justification be

given for this choice? Alternatively, if this variation in this

parameter is just a robustness check, and nothing of theoretical

interest, perhaps state this clearly.

Minor issues

- section “Sequential Bayesian Learning over Blocks and Tasks”. Which

model is used here? I’m assuming the restricted capacity model, but

this should be stated clearly.

- Can Figure 6 and 7 be combined in a single figure, so its easier to

judge the closeness of behaviour and model predictions?

- line 576. Subjects may not have been told that they would be tested

again on Task 1, but they were also not told the contrary. Why would

they assume they would not encounter the first task again?

- line 628: “non-linearly separable categories are easier to learn.”

Easier than what? And do the authors mean “linearly separable

categories”?

- Appendix C.2: How was it determined that an open-ended self-report

of a participants’ strategy matched their actual strategy? This

would be quite a complicated thing to determine, so more information

on the procedure is needed.

**Have all data underlying the figures and results presented in the manuscript been provided?**

Reviewer #2: None

Reviewer #3: Yes

PLOS authors have the option to publish the peer review history of their article (what does this mean?). If published, this will include your full peer review and any attached files.

Reviewer #2: No

Reviewer #3: No
---

## [Decision Letter · Decision Letter 2]

18 May 2021

Dear Dr Menghi,

We are pleased to inform you that your manuscript 'Multitask Learning over Shared Subspaces' has been provisionally accepted for publication in PLOS Computational Biology. Please see the small minor comment remaining below.

Best regards,

Samuel J. Gershman

Deputy Editor

PLOS Computational Biology

Reviewer's Responses to Questions

**Comments to the Authors:**

Reviewer #2: I don't have additional comments, thank you to the authors for the revision.

Reviewer #3: Review of “Multitask Learning over Shared Subspaces”

Summary

This study presents an experiment on transfer between tasks with a

shared or non-shared feature subspace, as well as three artificial

neural network models with Sequential Bayesian Learning. The experiment

shows a benefit in performance when the initial task contains a

subtraction structure and the second task shares this subspace, and the

modelling results show that a “minimal capacity” ANN with SBL matches

patterns in human performance better than an “increased capacity” ANN

and a model with “reduced precision” (which increases the variance of

the prior in the transfer task).

Evaluation

I’m satisfied with this second revision. The authors took my comments

(and those of the other reviewer) on board responded well to them. One

final thing is that the abstract seems not entirely reflective of the

more nuanced results of the present version:

In the abstract, it states “We found, as hypothesised, that subject

performance was significantly higher on the second task if it shared the

same subspace as the first. Additionally, accuracy was positively

correlated over subjects learning same-subspace tasks, and negatively

correlated for those learning different-subspace tasks. Additionally,

accuracy was positively correlated over subjects learning same-subspace

tasks, and negatively correlated for those learning different-subspace

tasks.”. This is not an accurate reflection of the results of the new

analyses. Regarding accuracy, the increase is higher for the same vs

different Task 2 only in the Task1=Sub1 condition. The hypothesised

pattern does not hold in general. Regarding correlations, overall, there

is significant positive correlation when the second task is the same.

The correlation is not significant when the task is different. And when

analysing separately by Task 1 subspace, the positive correlation is

only found in the Add/Add condition; other correlations are not

significant. So the statement about the negative correlation should be

removed.

This issue should be easy to resolve.

**Have the authors made all data and (if applicable) computational code underlying the findings in their manuscript fully available?**

Reviewer #2: None

Reviewer #3: Yes

PLOS authors have the option to publish the peer review history of their article (what does this mean?). If published, this will include your full peer review and any attached files.

Reviewer #2: No

Reviewer #3: **Yes: **Maarten Speekenbrink

---

## [Editor Report · Acceptance letter]

28 Jun 2021

PCOMPBIOL-D-20-01352R2 

Multitask Learning over Shared Subspaces

Dear Dr Menghi,

I am pleased to inform you that your manuscript has been formally accepted for publication in PLOS Computational Biology. Your manuscript is now with our production department and you will be notified of the publication date in due course.

With kind regards,

Agota Szep
